# Hemoglobin Binding to the Red Blood Cell (RBC) Membrane Is Associated with Decreased Cell Deformability

**DOI:** 10.3390/ijms25115814

**Published:** 2024-05-27

**Authors:** Gregory Barshtein, Leonid Livshits, Alexander Gural, Dan Arbell, Refael Barkan, Ivana Pajic-Lijakovic, Saul Yedgar

**Affiliations:** 1Department of Biochemistry, The Faculty of Medicine, The Hebrew University of Jerusalem, Jerusalem 9112001, Israel; saulye@ekmd.huji.ac.il; 2Red Blood Cell Research Group, Vetsuisse Faculty, Institute of Veterinary Physiology, University of Zurich, 8057 Zürich, Switzerland; leonidlivshts@gmail.com; 3Blood Bank, Hadassah University Hospital, Jerusalem 9112001, Israel; gural@hadassah.org.il; 4Pediatric Surgery, Hadassah University Hospital, Jerusalem 9112001, Israel; arbell@hadassah.org.il; 5Department of Digital Medical Technologies, Holon Institute of Technology, Holon 5810201, Israel; refaelb@hit.ac.il; 6Department of Chemical Engineering, University of Belgrade, 11000 Belgrade, Serbia; iva@tmf.bg.ac.rs

**Keywords:** red blood cells, RBC deformability, hemoglobin, membrane-bound hemoglobin

## Abstract

The deformability of red blood cells (RBCs), expressing their ability to change their shape as a function of flow-induced shear stress, allows them to optimize oxygen delivery to the tissues and minimize their resistance to flow, especially in microcirculation. During physiological aging and blood storage, or under external stimulations, RBCs undergo metabolic and structural alterations, one of which is hemoglobin (Hb) redistribution between the cytosol and the membrane. Consequently, part of the Hb may attach to the cell membrane, and although this process is reversible, the increase in membrane-bound Hb (MBHb) can affect the cell’s mechanical properties and deformability in particular. In the present study, we examined the correlation between the MBHb levels, determined by mass spectroscopy, and the cell deformability, determined by image analysis. Six hemoglobin subunits were found attached to the RBC membranes. The cell deformability was negatively correlated with the level of four subunits, with a highly significant inter-correlation between them. These data suggest that the decrease in RBC deformability results from Hb redistribution between the cytosol and the cell membrane and the respective Hb interaction with the cell membrane.

## 1. Introduction

Red blood cells (RBCs) have unique mechanical properties, which are essential for their optimal function [1,2,3]. A major flow-affecting property is the RBC deformability, expressing their ability to change shape as a function of flow-induced shear stress. This enables them to optimize oxygen delivery to the tissues and minimize their resistance to flow, especially in the microcirculation. Accordingly, impaired RBC deformability has been commonly implicated in the pathophysiology of conditions related to blood circulation disorders (e.g., cardiovascular diseases, diabetes, hemoglobinopathies, etc.).

RBC deformability is determined by various factors, including (a) the surface area-to-volume ratio of the cell [4] and (b) the membrane viscoelasticity [5], both regulated by the membrane structure and composition, as well as (c) intracellular viscosity [6]. Accordingly, any infringement of the cell membrane is expected to affect the cell deformability.

Hemoglobin (Hb) is the main protein of red blood cells and is a two-way gas carrier, transporting oxygen from the lungs to the tissues and carbon dioxide back. Each mature human red blood cell contains ~250–270 million copies of hemoglobin molecules [7]. Typically, in the red blood cells of a healthy adult, α (HBA) and β-subunits (HBB) exist in approximately 200 million copies each [7]. In addition, the less common δ (HBD) and γ (HBG) subunits were found in approximately 30 and 4 million copies, respectively. It was also reported [7] that significant amounts of θ-subunit (HBQ) (400,000 copies), zeta (HBZ) (900,000 copies), and mu (HBM) (100,000 copies) subunits can be found in the RBC cytoplasm in adults. However, these subunits are generally considered early embryonic forms of hemoglobin.

Intracellular hemoglobin concentrations (MCHC) in healthy subjects ranged from 32 to 36 g/dL [8]. This extremely concentrated protein solution is highly viscous [9] and exhibits inhomogeneous distribution in the cytosol [10,11,12]. In normal RBCs, the main part of Hb is dissolved in the cytosol, while 2–10% is bound to the inner membrane layer [13,14,15,16] and makes up about 20% of the membrane mass [17]. The attachment of Hb to the RBC’s membrane’s inner surface [18,19,20] and the enhanced hemoglobin–membrane association in sickle RBCs [21,22,23] have been detected by analysis of the membrane composition of RBC ghosts. This was further found in intact RBCs in a Raman spectroscopy study, showing cytosolic and membrane-bound hemoglobin [10,11,12] and elevation of deoxygenated Hb (deoxy-Hb) near the membrane surface [24]. The binding of Hb to the RBC membrane is involved in the aging of adult and neonatal RBC and subsequent cell removal from circulation by phagocytosis [12,25].

The mechanism of hemoglobin interaction with cell membranes has been discussed in detail in the literature but still needs to be fully understood. A major factor in this process is band-3 [26], which preferentially binds deoxy-Hb [27,28,29,30]. In physiological conditions, less than 1% of Hb can be oxidized to form methemoglobin (metHb) and FerriHb, which can also bind to the normal RBC membrane [31,32]. However, as their concentration in the cytosol is less than 1% [33,34], their presence on the membrane is insignificant.

The membrane-bound hemoglobin (MBHb) affects the functionality of the membrane and the entire cell [35]: it modulates glycolysis [35,36] and oxygen delivery to tissues; promotes senescence and the consequent cell targeting for removal from circulation via phagocytosis by macrophages [33]; disrupts several RBC characteristics, including cell surface charge, membrane permeability, and mechanical properties [12,37]; increases membrane shear viscosity [38] and time constants for cell shape recovery [16] but does not change the membrane elasticity [38]; and induces thickening of the membrane and makes the inner leaflet rougher than the outer leaflet [23].

The above findings suggest that Hb interaction with the RBC membrane can affect the cell’s mechanical properties, particularly deformability. A previous study indicated this, showing that Hb interaction with the RBC membrane is associated with decreased deformability [39]. However, until now, no quantitative relationship has been established between the amount of MBHb and cell deformation. In addition, the relation of the different MBHb isoforms to the alteration of the cell deformability has not been analyzed.

On these grounds, the present study was undertaken to comprehensively explore the correlation between the bindings of various isoforms of hemoglobin and cell deformability. To this end, we analyzed the correlation of the membrane-bound Hb content, determined by mass spectroscopy (MS), with the cell deformability. The decrease in RBC deformability was found to be associated with the elevation of MBHb content on the cell membrane.

## 2. Results

### 2.1. Membrane-Bound Hemoglobin Subunits

Six hemoglobin subunits were found on the RBC membrane fraction. As depicted in Table 1, two chains, α and β, are most abundant. In addition, subunits of embryonic hemoglobin, µ, and θ-subunits were also identified, although the content of θ chain is near the detection limit.

### 2.2. Correlations between the Levels of Membrane-Bound Hb Subunits

The levels of membrane-bound Hb subunits can express independent binding for the individual proteins or be interdependent. To explore this, we analyzed the inter-correlation between the levels of Hb subunits which constitute the three main Hb isoforms in adult RBCs, HbA0 (α2β2), HbA2 (β2δ2), and HbF (β2γ2). The correlation matrix in Table 2 shows that highly significant inter-correlations between α, β, and δ subunits were found. All correlations were positive, implying inert dependency between the binding of these proteins and the respective HbA0 and HbA2 to the RBC membrane. A significant correlation was observed for the contents of embryonic Hb forms, μ, and θ. In turn, no significant correlation was found between the content of α, β, and δ subunits and that of γ, μ, and θ-subunits.

### 2.3. Variability in the Level of the Membrane-Bound Hb Subunits

The SD values shown in Table 1 indicate large variability in the levels of the membrane-bound Hb subunits. Figure 1 illustrates this, showing the variability in the subunit content specified in Table 2.

Due to solid donor-to-donor and unit-to-unit variability, we did not observe a difference between subunit β, α, δ, μ, and θ in freshly collected and stored samples, while clear elevation of the level of γ-subunit content was observed in stored RBCs, as shown in Table 3.

### 2.4. Deformability of RBCs

As described in our previous study [40], no significant difference was found in the deformability of fresh and stored RBCs, expressed by the average elongation ratio (AER), with AER = 1.58 ± 0.09 and 1.54 ± 0.12, respectively.

### 2.5. Correlations between the Content of the Membrane-Bound Hb Subunits and the RBC Deformability

The levels of the six membrane-bound Hb subunits were examined for correlation with RBC deformability, as measured by the flow-induced cell elongation and expressed by the average elongation ratio (AER, see Section 4.2.6).

Table 4 shows a statistically significant negative correlation between cell deformability and the levels of the membrane-bound β, α, and δ subunits; cell deformability decreases with increasing content of the membrane-bound Hb subunits. Figure 2 elaborates on this, showing the inverse relationship between the membrane-bound β, α, and δ subunit level and RBC deformability (AER). In contrast, no significant correlation was found between the membrane-bound μ, θ, and γ subunit content and cell deformability.

## 3. Discussion

As noted above, red blood cell deformability is a critical factor in oxygen delivery to tissues and blood flow, especially in the microcirculation, and impaired RBC deformability has been commonly implicated in the pathophysiology of conditions related to blood circulation disorders [41]. The deformability is determined by various factors, primarily the cell membrane structure and composition [40,42] and intracellular Hb concentration (MCHC value) [43,44,45]. These include the structural organization of the cytoskeleton and proteins and the state of the bilayer lipid distribution. In particular, a decrease in deformability (increased rigidity) of an RBC is exerted by changes in band-3 distribution along the membrane and its clustering [46], reduction of spectrin flexibility [47], and translocation of phosphatidylserine to the cell surface [48]. Accordingly, interactions that affect the RBC membrane structure and composition, especially the factors mentioned above, would affect cell deformability.

The binding of hemoglobin with the inner leaflet of the RBC membrane is a common, long-known phenomenon [49] and has been the subject of numerous studies [10,11,12]. Recently, Livshits et al. showed that Hb association with the cell membrane is reversible and regulated by the level of intracellular Ca^2+^ [50]. Welbourn et al. [33] presented a model that oxidative stress triggers Hb attachment to RBC membranes throughout the cell’s life cycle, and this becomes increasingly significant as the cell ages. Specifically, several authors, under the condition of light oxidation stress, demonstrated that Hb (both α and β hemoglobin) binds in a high molecular weight complex containing band 4.2, ankyrin and spectrin that is directly associated with band-3 of the erythrocyte membrane [51,52,53].

However, the effect of the membrane-bound Hb on RBC deformability has been addressed in only one study, which was reported decades ago [39], which reported that Hb interaction with the RBC membrane is associated with decreased deformability. A recent study [54] reported that the detachment of Hb from the RBC membrane was associated with increased deformability. In contrast, another study with cats with methemoglobinemia reported that the binding of MetHb to the RBC membrane induces elevation of cell deformability.

For the first time, the present study provides a quantitative analysis of the binding of the various Hb isoforms to the RBC membrane and its effect on cell deformability. As shown in Table 1, six Hb subunits, which have been previously identified in RBC lysates [7], were found attached to the isolated RBC membrane. In the present study, we focused on the four most abundant in adult RBCs. Table 2 shows a clear inter-correlation between the content of the membrane-bound α, β, and δ subunits but not for the γ, μ, and θ subunits, which are characteristic for fetal and embryonic forms of hemoglobin, correspondently. This difference can be due to the non-homogenous distribution of fetal Hb (HbF) in adult RBCs, predominating in a subset of cells termed “F cells” [55,56], and relatively minor expression of the embryonic forms.

In contrast, HbF, HbA, and HbA2 are ubiquitously distributed in adults’ RBCs [57]. The inter-correlation between their membrane-bound levels (Table 2) suggests that their binding to the membrane is not competitive, and the limiting factor in this process is the membrane’s ability to bind specific Hb molecules. Accordingly, as expected, the prominent membrane-bound isoform was HbA (Table 1), which is the most abundant in the cytosol and, due to its positive charge, has a higher affinity to RBC membrane proteins than other Hb isoforms [21]. The ratio between the membrane-bound δ and α-subunits is 0.042 ± 0.015, resembling the range observed by Livshits et al. [50] for HbA2/HbA. MetHb can also bind to the membrane, but due to its low concentration in the cytosol of normal RBCs, its presence can be neglected [33,34].

In the present study, strong inverse correlations were found between the cell deformability and the levels of the membrane-bound Hb subunits β, α, and δ but not for the γ or embryonic subunits (Table 4). The mechanism by which MBHb affects RBC deformability has not been studied directly, but the Hb interactions with the RBC membrane components, which have been studied extensively [19,27,51,58], may explain this effect, as follows:

Three main factors determine the RBC deformability: (a) the surface area-to-volume ratio of a cell [4], (b) the membrane viscoelasticity [5], and (c) intracellular viscosity [6]. The relationship between these factors and the binding of hemoglobin to the membrane still needs to be better studied, and only scattered information on this issue is provided in the literature. Yet, it was found that binding Hb to the membrane provokes an increase in the membrane viscosity but does not change the membrane elasticity [38].

Hb molecules interact with membrane proteins, such as α- and β-spectrin, actin, and protein 4.1 [59]. Thus, Hb attachment to spectrin molecules causes a reduction in the spectrin flexibility, which has a feedback impact on the viscoelasticity of the actin–spectrin cortex, enhancing its rigidity. Band-3 is probably the most critical target for Hb membranous binding [19,27,33]. The electrostatic binding of Hb molecules to the cytoplasmic domain of band-3, accompanied by the already mentioned protein–lipid positive hydrophobic mismatch effects, causes the band-3 conformation changes and the ability to diffuse and form clusters [60]. In turn, a change in the conformational state of band-3 molecules, their clustering, and space distribution can be considered one of the main factors that regulated the viscoelasticity of the cortex and the lipid bilayer, as was shown in aged RBCs [12,61,62]. Specifically, binding Hb molecules to band-3 may induce cortex stiffening. Moreover, some models have been suggested whereby the binding of Hb to the membrane, with the integrity of the cytoskeleton, results in band-3 clustering during RBC aging [63]. Thus, band-3 alterations affect the binding between the cytoskeleton and lipid bilayer and the signaling of ATP [62], subsequently worsening RBC deformability.

The spatial distribution of band-3 molecules and their lateral diffusion, accompanied by the attachment of Hb to the lipid bilayer, influences the bending modulus of the lipid bilayer [64,65,66]. The interaction of the actin cortex with the lipid bilayer can lead, depending on temperature, to either the formation of highly dynamic membrane domains (rafts) [40] or the prevention of large-scale phase separation [67]. Local changes in the bilayer bending state enhance anomalous diffusion in small loci, eventually leading to hop-diffusion of lipids. The bilayer structural changes also have a feedback impact on band-3 protein–lipid positive hydrophobic mismatch effects [68,69,70].

Notably, Koshkaryev et al. [48] specifically showed that band-3 clustering induced the translocation of phosphatidylserine to the RBC surface, causing increased RBC rigidity (reduced deformability), which was specifically expressed by the increased percentage of non-deformable cells [71].

In conclusion, we can assume the presence of the following cause-and-effect relationship: the presence of hemoglobin in the cell membrane induces a cascade of transformations, including band-3 clustering, a change in the state of the skeleton, and a decrease in cell deformation. In parallel, the clustering of band-3 can provoke the release of phosphatidylserine onto the outer surface of erythrocytes [48] and, thus, cause an increase in the presence of non-deformable cells [71].

Limitation of the study: as described in the Methods section, we used blood samples from young (18–37 years), nonsmoker healthy men with blood type O+ to minimize the heterogeneity of the subject population and potential side factors that can affect RBC deformability [17,72,73,74,75].

## 4. Materials and Methods

### 4.1. Materials

All chemicals were purchased from Sigma Aldrich Israel (Jerusalem, Israel), except for Ca^2+^/Mg^2+^-free phosphate-buffered saline (PBS) (Biological Industries, Beit-Haemek, Israel).

### 4.2. Methods

#### 4.2.1. RBC Sample Sources

As in a previous study [40], blood samples were collected from young, nonsmoker, healthy male volunteers (18–37 years) with blood type O+ to obtain a homogeneous population. This selection of volunteers was made to minimize the influence of external factors on the interaction of hemoglobin with the membrane and the cell deformability [17,72,73,74,75].

RBCs were obtained from two sources:Nine blood samples were collected from nonsmoker healthy male donors (18–37 years, group O+, Hb > 13 mg/dL) without known disorders;Blood was collected from six nonsmoker healthy donors (group O+) per the blood bank routine and stored in sterile bags in SAGM at 4 °C in the Hadassah Hospital Blood Bank until their expiration date (42 days).

All blood samples were taken under the approval of the Helsinki Ethics Committee of the Hadassah Hospital, Jerusalem, Israel. Permit 0819-20-HMO.

#### 4.2.2. Isolation of RBCs from Freshly Collected Blood

Blood was drawn into EDTA-containing tubes. RBCs were isolated and washed twice in PBS by centrifugation (500× *g* for 10 min) and re-suspended in the same buffer supplemented with 0.5% BSA.

#### 4.2.3. Packed RBCs (PRBC)

RBCs were collected from packed RBC units and, as above, washed and re-suspended in Ca^2+^/Mg^2+^-free PBS supplemented by 0.5% of BSA.

#### 4.2.4. Preparation of RBC Membranes

Cells were lysed in lysis buffer (5 mM sodium phosphate pH 8.0, 1 mM EGTA, 1 mM EDTA, and 1 mM PMSF) and placed on ice for 10 min. The hemolyzed RBC suspension was centrifuged (14,000 rpm for 10 min), and the pellet was washed three times to obtain clean RBC membranes, re-suspended in the buffer, and supplemented with a protease inhibitor cocktail (Sigma, St. Louis, MO, USA). Samples were sent in dry ice for mass spectrometry analysis.

#### 4.2.5. Determination of MBHb Composition

The composition of the membrane-bound Hb subunits and their levels were comprehensively determined using mass spectroscopy, using label-free quantification (LFQ) to determine the protein content relative to a standard reference [76,77]. Proteins were trypsin-digested following the in-solution digestion protocol. Peptides were then purified on C18 StageTips (Fisher Scientific, Hampton, NH, USA) before their LC-MS analysis. Peptides were separated on an Easy-spray PepMap column (Thermo Scientific Fisher, Waltham, MA, USA) using a water–acetonitrile gradient and the EASY-nLC1000 nanoHPLC. Peptides (Thermo Scientific Fisher, Waltham, MA, USA) were electrosprayed into a Q-Exactive mass spectrometer (Thermo Scientific Fisher, Waltham, MA, USA) via the Easy-spray source. Peptides were analyzed using a data-dependent acquisition, with the fragmentation of the top 10 proteins from each scan. Raw MS files were analyzed by MaxQuant (Max Planck Institute of Biochemistry, Munich, Germany) using the Human Uniprot database. The false discovery rate was set to 1% at the protein and peptide levels. Mass spectrometry identified and quantified six Hb subunits and their level.

#### 4.2.6. Determination of RBC Deformability

RBC deformability was determined using our computerized cell flow-properties analyzer (CFA), as described in our previous studies [2,40,50,71] and illustrated by Figure 3. RBC deformability is measured by directly visualizing the change in the RBC shape in a narrow-gap flow chamber under flow-induced shear stress, expressing the conditions in microvessels.

Then, 50 µL of the RBC suspension (1% hematocrit) in PBS buffer, supplemented with 0.5% BSA, is inserted into the flow chamber containing a glass slide, to which the RBC adheres. After 15 min of incubation, the adherent RBCs are subjected to flow-induced shear stress of 3.0 Pa (for illustration, see Figure 4). Using image analysis, the deformability is determined for each cell by its elongation and expressed by the elongation ratio, ER = a/b, where “a” is the major cellular axis, and “b” is the minor cellular axis, where the lowest value of ER is 1.0 reflecting a round, un-deformed RBC. The average ER (AER) was derived from the ER values of 8000–10,000 cells from 20 to 25 images.

#### 4.2.7. Statistical Analysis

The Shapiro–Wilk test was used to verify the normality of the distribution of the continuous variables. The Pearson coefficient and *p* value characterized the significance of linear regression between two tested parameters.

## 5. Conclusions

This study presents, for the first time, a quantitative exploration of the correlation between RBC deformability (using image analysis) and the levels of membrane-bound hemoglobin (MBHb) subunits (using mass spectroscopy). Six MBHb subunits were identified. Highly significant correlations were found between RBC deformability and the levels of three MBHb subunits β, α, and δ. A strong inter-correlation was found between the levels of these MBHb subunits. This study suggests that the decrease in RBC deformability results from Hb redistribution between the cytosol and the cell membrane, as well as the interactions of the respective membrane-bound Hb subunits with the cell membrane.

## Figures and Tables

**Figure 1 ijms-25-05814-f001:**
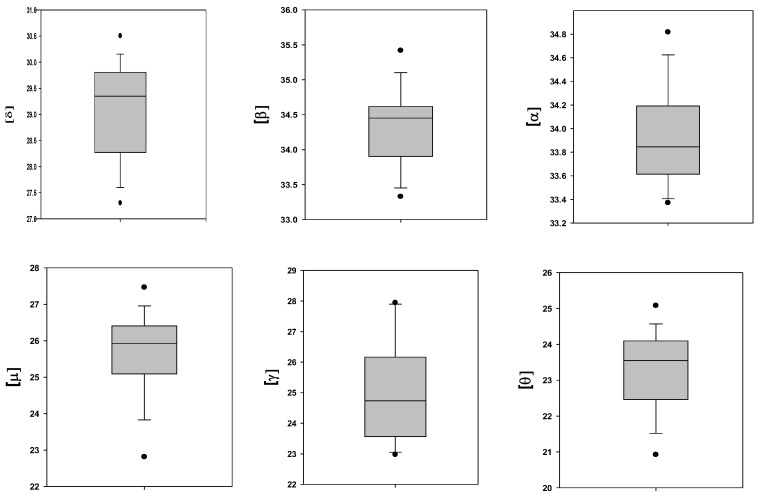
Variability in the content of membrane-bound Hb-subunits (expressed by (Ln (LFQ)). Statistical analysis was carried out for 15 samples of healthy adult RBCs.

**Figure 2 ijms-25-05814-f002:**
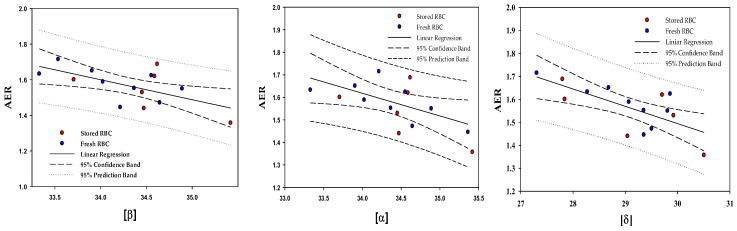
Correlation between the level of the RBC membrane-bound β, α, and δ subunits (Ln (LFQ)) and the cell deformability (AER); for statistical analysis, see Table 4.

**Figure 3 ijms-25-05814-f003:**
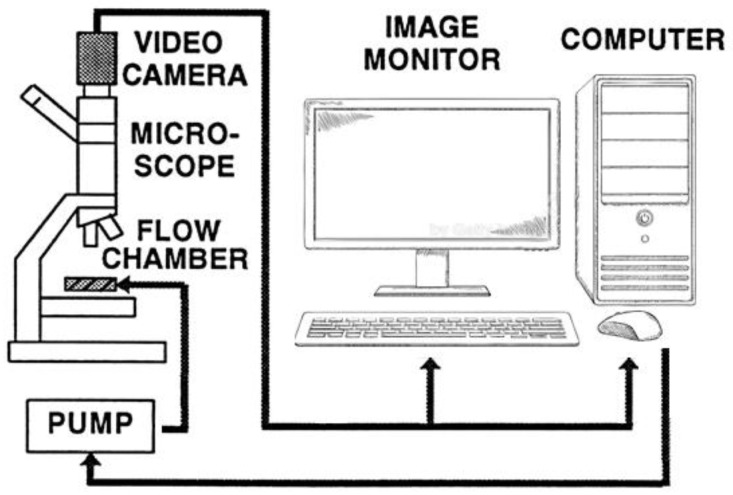
Scheme of cell-flow analyzer CFA.

**Figure 4 ijms-25-05814-f004:**
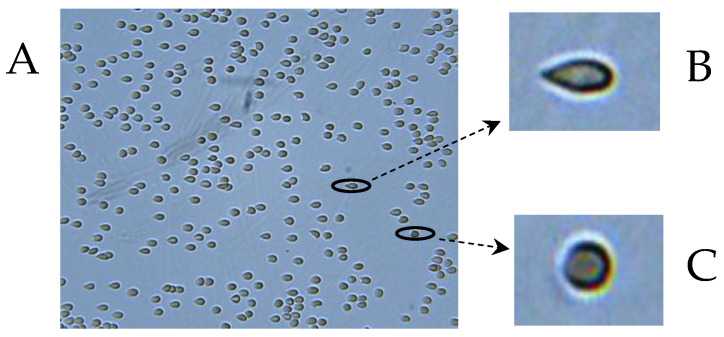
A. Image of RBC field in the CFA flow-chamber under flow-induced shear stress of 3.0 Pa. B. Highly deformable cell (ER = 2.3). C. Non-deformable cell (ER = 1).

**Table 1 ijms-25-05814-t001:** The content of the hemoglobin subunits bound to the RBC membrane (expressed by Ln (LFQ)). Each datum is mean ± SD for 15 samples of healthy adult RBCs.

Number	Hemoglobin Subunit	Gene	Content; Ln(LFQ)
1	β	*HBB*	34.4 ± 0.5
2	α	*HBA1*	34.0 ± 0.4
3	δ	*HBD*	29.1 ± 0.9
4	µ	*HBM*	25.7 ± 1.1
5	γ	*HBG2*	25.1 ± 1.6
6	θ	*HBQ1*	23.3 ± 1.1

**Table 2 ijms-25-05814-t002:** Correlation matrix for the levels of the membrane-bound Hb subunits. Statistical significance is expressed using the Pearson correlation coefficient. Statistical analysis was carried out for 15 samples of healthy adult RBCs. *p* value is *- < 0.05; **- ≤ 0.001.

	β	α	δ	μ	γ	θ
β	1.00					
α	0.82 **	1.00				
δ	0.76 **	0.76 **	1.00			
μ	0.25	0.07	0.63	1.00		
γ	0.37	0.08	0.43	0.21	1.00	
θ	0.42	0.46	0.39	0.54 *	0.26	1.00

**Table 3 ijms-25-05814-t003:** Levels of MBHb subunits in freshly donated vs. stored RBCs.

№	Hemoglobin Subunit	Content; Ln(LFQ)		Significance, *p*
		Fresh	Stored	
1	β	34.2 ± 0.5	34.5 ± 0.6	0.20
2	α	34.1 ± 0.4	33.9 ± 0.2	0.31
3	δ	29.0 ± 0.8	29.1 ± 1.1	0.84
4	μ	26.0 ± 0.6	25.2 ± 1.4	0.09
5	γ	24.4 ± 0.5	26.7 ± 0.8	0.007
6	θ	23.4 ± 0.7	23.4 ± 1.5	0.45

**Table 4 ijms-25-05814-t004:** Correlations between the levels of membrane-bound Hb subunits, expressed by Ln (LFQ), and the RBC deformability, expressed by the average elongation ratio (AER). NS—non-significant.

№	Hemoglobin Subunit	Significance, *p*	Pearson Coefficient, *r*
1	β	0.017	−0.606
2	α	0.0053	−0.722
3	δ	0.0047	−0.687
4	γ	NS	-
5	μ	NS	-
6	θ	NS	-

## Data Availability

Data are contained within the article. The data presented in this study are available upon request from the corresponding author.

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
