# Peer review of "Hemoglobin Binding to the Red Blood Cell (RBC) Membrane Is Associated with Decreased Cell Deformability"

_ijms, 2024, doi:10.3390/ijms25115814_

Round 1

Reviewer 1 Report (New Reviewer)

Comments and Suggestions for Authors

In the present study, Barshtein et al. investigated the effect of haemoglobin binding to the RBC membrane on cell deformability. The authors analysed fresh and stored RBCs from healthy male donors regarding membrane-bound Hb and deformability. Six Hb subunits were identified to be bound to the RBC membrane and the authors state a negative correlation between membrane-bound Hb content and cell deformability.

Although in the methods section, the authors present two cohorts, including RBC from fresh blood and RBC after 42 days of storage in a blood bank, in the results section, there is no distinction between the two groups. Since it is known that RBC storage affects RBC deformability, it would be interesting, if the RBC storage did actually affect deformability in this study and if it inversely affected the content of membrane-bound Hb. Do the authors have data comparing stored and fresh RBCs regarding these two issues? Unfortunately, in the manuscript, data are only presented for the overall cohort.

Additional to the membrane-bound Hb, a comparison to the overall Hb content of the related sample would be interesting.

Fig.1: Could the authors please provide box and whisker plots for all Hb subunits to give an insight in the variability between the subunits regarding ranges?

Please provide a definition/explanation for Ln(LFQ).

Fig.2: It would be informative to have differently coloured dots for fresh and stored RBCs to see possible differences in the graph at first sight. Furthermore, please provide an axis-label (mainly x-axis) with corresponding units. It would be nice to have the same figures for all investigated Hb subunits, not only for Hb delta.

A picture of the computerized cell flow-properties analyser (CFA) and a representative picture of the deformation of RBCs in the flow cell would be nice to optically underline the results.

Altogether, the study is done properly using state of the art methods, although the study cohort with 9 and 6 donors per group is quite small. The graphical presentation of the results could be more extensive and catchy.

Author Response

Reviewer 1

In the present study, Barshtein et al. investigated the effect of haemoglobin binding to the RBC membrane on cell deformability. The authors analyzed fresh and stored RBCs from healthy male donors regarding membrane-bound Hb and deformability. Six Hb subunits were identified to be bound to the RBC membrane and the authors state a negative correlation between membrane-bound Hb content and cell deformability.

Although in the methods section, the authors present two cohorts, including RBC from fresh blood and RBC after 42 days of storage in a blood bank, in the results section, there is no distinction between the two groups. Since it is known that RBC storage affects RBC deformability, it would be interesting, if the RBC storage did actually affect deformability in this study and if it inversely affected the content of membrane-bound Hb.

Do the authors have data comparing stored and fresh RBCs regarding these two issues? Unfortunately, in the manuscript, data are only presented for the overall cohort.

Others and we [1-3] have addressed the effect of RBC storage on their deformability in numerous studies, showing that the deformability decreases during storage in some blood units, while it is not significantly affected in other units [2]. Similarly, in the present study, we did not observe a clear distinction between the deformability of freshly-collected and outdated stored RBC, as well as between the respective membrane-bound hemoglobin subunit (MBHb), as shown in Table 3. Following the reviewer's comment, we added this information to the revised manuscript (see lines 134-136 and 145-147).

 Additional to the membrane-bound Hb, a comparison to the overall Hb content of the related sample would be interesting.

Several studies have reported that increased MCHC (intracellular Hb content) is associated with decreased cell deformability [4-6]. In the presented study, we focused on the membrane-bound Hb and provided, for the first time, a quantitative analysis of the correlation between various isoforms/subunits of the MBHb and the cell deformability.  

This point is further elaborated in a paragraph added to the revised manuscript (Line 190-191)

 Fig.1: Could the authors please provide box and whisker plots for all Hb subunits to give an insight into the variability between the subunits regarding ranges?

As recommended by the reviewer, the requested data was added to the revised Fig. 1.

Please provide a definition/explanation for Ln(LFQ).

The requested definition was added as requested (Line 317-318)

Fig.2: It would be informative to have differently colored dots for fresh and stored RBCs to see possible differences in the graph at first sight.

As recommended by the reviewer, the requested modification was made in Figure 2.

 Furthermore, please provide an axis-label (mainly x-axis) with corresponding units. It would be nice to have the same figures for all investigated Hb subunits, not only for Hb delta.

Fig. 2 was modified as recommended by the reviewer.

A picture of the computerized cell flow-properties analyzer (CFA) and a representative picture of the deformation of RBCs in the flow cell would be nice to optically underline the results.

As recommended by the reviewer, the revised manuscript included a schema of CFA and images of RBC (Figs 3 & 4) in the flow chamber.

Overall, the study is done properly using state-of-the-art methods, although the study cohort, with 9 and 6 donors per group, is quite small. The graphical presentation of the results could be more extensive and catchy.

We believe that the changes made in the revised manuscript satisfactorily address the reviewer’s critique and sincerely thank him for his constructive comments and suggestions.

References

  1. Barshtein, G.; Rasmusen, T.L.; Zelig, O.; Arbell, D.; Yedgar, S. Inter-donor variability in deformability of red blood cells in blood units. Transfus Med 2020, 30, 492-496, doi:10.1111/tme.12725.
  2. Barshtein, G.; Gural, A.; Zelig, O.; Arbell, D.; Yedgar, S. Unit-to-unit variability in the deformability of red blood cells. Transfus Apher Sci 2020, 59, 102876, doi:10.1016/j.transci.2020.102876.
  3. Barshtein, G.; Gural, A.; Zelig, O.; Arbell, D.; Yedgar, S. Preparation of packed red blood cell units in the blood bank: Alteration in red blood cell deformability. Transfus Apher Sci 2020, 59, 102738, doi:10.1016/j.transci.2020.102738.
  4. Bosch, F.H.; Werre, J.M.; Schipper, L.; Roerdinkholder-Stoelwinder, B.; Huls, T.; Willekens, F.L.; Wichers, G.; Halie, M.R. Determinants of red blood cell deformability in relation to cell age. Eur J Haematol 1994, 52, 35-41, doi:10.1111/j.1600-0609.1994.tb01282.x.
  5. Lutz, H.U.; Stammler, P.; Fasler, S.; Ingold, M.; Fehr, J. Density separation of human red blood cells on self forming Percoll gradients: correlation with cell age. Biochim Biophys Acta 1992, 1116, 1-10, doi:10.1016/0304-4165(92)90120-j.
  6. McNamee, A.P.; Richardson, K.; Horobin, J.; Kuck, L.; Simmonds, M.J. Susceptibility of density-fractionated erythrocytes to subhaemolytic mechanical shear stress. Int J Artif Organs 2019, 42, 151-157, doi:10.1177/0391398818790334.

Reviewer 2 Report (New Reviewer)

Comments and Suggestions for Authors

The present manuscript presents only indirect, weakly linked and weakly presented data about the correlation of hemoglobin subunit binding and cell deformability. Differences are presented on gene level, while their organisation and proper binding is not presented, as the deformability is measured by cell shape, not by direct mechanical indentation. Conclusion contains mostly speculation and hypothesis, which is admitted in text not being proved by the results. I consider the manuscript having low novelty and scientific importance. Might be reconsidered after major revision. (Direct measurements included).

Author Response

Reviewer 2

The present manuscript presents only indirect, weakly linked and weakly presented data about the correlation of hemoglobin subunit binding and cell deformability.

We are sorry to disagree with the reviewer's opinion, as we find his comments unclear (as long as we could understand them), or wrong…

 ….only indirect… data…- In the present study, we determined (1) the MBHb subunits using the conventional, commonly used mass spectroscopy and (2) the RBC deformability by direct visualization of the cell's shape change under flow-induced shear stress (the exact definition of the cell deformability). What could be more direct than that?

…. weakly linked…..- The link between the MBHb and cell deformability was analyzed using the conventional statistical method, showing highly significant correlations between the levels of Hb subunits β, α, and δ and the cell deformability. In contrast, no correlation was found with Hb γ (Table 3). We thus do not really understand what the reviewer meant by "weakly linked."

…..weakly presented data….As for the comment above, we don't really understand what the reviewer meant. The data is presented straightforwardly, and we will be happy to learn if the reviewer has any suggestions for better presentation.

Differences are presented on the gene level, while their organization and proper binding are not presented, as deformability is measured by cell shape, not by direct mechanical indentation.

 Differences are presented on the gene level, while their organization and proper binding are not presented

Gene level is the conventional expression in mass spectroscopy;

We don’t understand what difference the reviewer is asking about (between what and what?);

The MBHb was determined in isolated RBC membranes. What does he mean by “proper binding"? 

…….. the deformability is measured by cell shape, not by direct mechanical indentation.

 By definition, RBC deformability is the ability of the cell to change its shape under flow-induced shear stress [1-7]. Several methods are used to determine RBC deformability, mostly indirect (filtration, viscosimetry, and ektacytometry). As shown in our numerous previous studies and illustrated in Figs 3 & 4 of the revised manuscript, we determine RBC deformability by direct visualization (under a microscope) of the cell shape change under flow-induced shear stress, which exactly corresponds with the definition of RBC deformability.

Notably, the membrane “mechanical indentation” (mentioned by the reviewer) is not identical to the whole cell deformability.

Conclusion contains mostly speculation and hypothesis, which is admitted in text not being proved by the results.

 The reviewer's comment was taken into account in the revised Conclusion.

I consider the manuscript having low novelty and scientific importance.

 Re: Novelty - Previous studies have reported QUALITATIVE data showing that an increase in total Hb binding to the RBC membrane leads to a rise in cell rigidity. In the present study, we present, for the first time, a QUANTITATIVE relationship between the level of membrane-bound hemoglobin subunits and cell deformability for the first time, showing

1 . A clear inverse correlation between the deformability of adults’ RBC and the level of the three main hemoglobin subunits, β, α, and δ, that bind to the inner surface of the cell membrane; 

  1. A strong inter-correlation between the levels of these membrane-bound subunits
  2. The binding of the different hemoglobin isoforms to the RBC membranes is not competitive.

Respectively, these points are specified in the revised "Conclusion" section (Line 362-369).

Might be reconsidered after major revision. (Direct measurements included).

Since the reviewer did not provide any specific constructive suggestions, we sincerely appreciate the other reviewers' constructive comments and revised the manuscript accordingly.

References

1. Barshtein, G.; Gural, A.; Arbell, D.; Barkan, R.; Livshits, L.; Pajic-Lijakovic, I.; Yedgar, S. Red Blood Cell Deformability Is Expressed by a Set of Interrelated Membrane Proteins. Int J Mol Sci 2023, 24, doi:10.3390/ijms241612755.

2. Arbell, D.; Bin-Nun, A.; Zugayar, D.; Eventov-Friedman, S.; Chepel, N.; Srebnik, N.; Hamerman, C.; Wexler, T.L.R.; Barshtein, G.; Yedgar, S. Deformability of cord blood vs. newborns' red blood cells: implication for blood transfusion. J Matern Fetal Neonatal Med 2022, 35, 3270-3275, doi:10.1080/14767058.2020.1818203.

3. Barshtein, G.; Pajic-Lijakovic, I.; Gural, A. Deformability of Stored Red Blood Cells. Front Physiol 2021, 12, 722896, doi:10.3389/fphys.2021.722896.

4. McVey, M.J.; Kuebler, W.M.; Orbach, A.; Arbell, D.; Zelig, O.; Barshtein, G.; Yedgar, S. Reduced deformability of stored red blood cells is associated with generation of extracellular vesicles. Transfus Apher Sci 2020, 59, 102851, doi:10.1016/j.transci.2020.102851.

5. Barshtein, G.; Rasmusen, T.L.; Zelig, O.; Arbell, D.; Yedgar, S. Inter-donor variability in deformability of red blood cells in blood units. Transfus Med 2020, 30, 492-496, doi:10.1111/tme.12725.

6. Barshtein, G.; Gural, A.; Zelig, O.; Arbell, D.; Yedgar, S. Unit-to-unit variability in the deformability of red blood cells. Transfus Apher Sci 2020, 59, 102876, doi:10.1016/j.transci.2020.102876.

7. Barshtein, G.; Gural, A.; Zelig, O.; Arbell, D.; Yedgar, S. Preparation of packed red blood cell units in the blood bank: Alteration in red blood cell deformability. Transfus Apher Sci 2020, 59, 102738, doi:10.1016/j.transci.2020.102738.

Reviewer 3 Report (New Reviewer)

Comments and Suggestions for Authors In this manuscript, the authors investigate the correlation between the content of membrane-bound haemoglobin in red blood cells and the deformability of the cells. They found a negative correlation between cell deformability and the four haemoglobin subunits as well as an intercorrelation between them.
In general, the research is sound and the article is well written, but to make it more attractive to the general public, it would be beneficial to explain the applicability of the results obtained here, e.g. for the treatment of diseases where red blood cell deformability is used for diagnosis.
In the conclusion, the authors should state what was done and what methods were used in the experiment, and omit speculations and hypotheses that are usually part of the discussion. Comments on the Quality of English Language

Author Response

Reviewer 3.

Comments and Suggestions for Authors

In this manuscript, the authors investigate the correlation between the content of membrane-bound haemoglobin in red blood cells and the deformability of the cells. They found a negative correlation between cell deformability and the four haemoglobin subunits as well as an intercorrelation between them.

In general, the research is sound and the article is well written, but to make it more attractive to the general public, it would be beneficial to explain the applicability of the results obtained here, e.g. for the treatment of diseases where red blood cell deformability is used for diagnosis.

 We are particularly grateful for the reviewer's comment on the clinical relevance of our findings. It is indeed tempting to speculate on their potential use in circulatory conditions, a topic we have extensively covered in our studies. However, we acknowledge that determining MBHb (using Mass Spectroscopy) is a laborious process, and it is still premature to make definitive statements about our findings' clinical or diagnostic applications.

 In the conclusion, the authors should state what was done and what methods were used in the experiment and omit speculations and hypotheses that are usually part of the discussion.

 The conclusion section was revised accordingly (see Lines 362-369).

We appreciate the reviewer’s constructive critique and sincerely thank him for that.

Round 2

Reviewer 1 Report (New Reviewer)

Comments and Suggestions for Authors

All comments were addressed sufficiently by the authors.

Author Response

We are grateful to the reviewer for his positive assessment of our work.

Reviewer 2 Report (New Reviewer)

Comments and Suggestions for Authors

The authors improved somewhat the manuscript, but I am not convinced about the mechanical deformability of the studied cells. One thing is how they appear in the microscope, another is how they react to mechanical stress (here I mean direct indentation measurement by e.g. atomic force microscopy, laser tweezers or other direct mechanical measurement). 

Author Response

Thanks to the reviewer for asking the question. We agree with the reviewer that each method for measuring cell deformation has its own specifics. During blood circulation, cells are exposed to shear stress created by the flow. The ability to deform and change the shape of a cell without stretching its membrane leads to a decrease in flow resistance and also allows cells to penetrate into capillaries. This deformation occurs solely due to the shear stress created by the flow being applied to the cells. This is precisely the situation that the setup we use in our device (see lines 330 - 357). As shown in Figure 4, cells are deformed under the influence of flow, and we evaluate this ability.

This manuscript is a resubmission of an earlier submission. The following is a list of the peer review reports and author responses from that submission.

Round 1

Reviewer 1 Report

Comments and Suggestions for Authors

Barshtein et al. demonstrate in a short communication that hemoglobin binding to RBCs correlates with lower cell deformability.

I have some major and minor comments that should be addressed before submission of this article.

Minor comments:

line 38: "(e.g., cardiovascular diseases, ...)

line 40: "cell [4] and (b) the ..." (spacer is missing)

line 47: reference missing

line 54: reference is missing

line 89-91: Sentence lacks words... "are most abandoned on the WHAT?" and do the authors mean "detection limit" instead of "detection level"?

line 124: I think it should be "flow-induced".

Major comments:

1. The authors used only RBCs from male volunteers. Is there a difference to RBCs from women.

2. Is there a difference between different blood groups?

3. To be able to systematically analyze the described issues, I think that one need to characterize the RBC deformability in a Hb concentration dependent manner - even with the subunits. Are the authors able to provide such analysis for at least HBA and HBB at once? This would serve as a defined proof of your study and would increase the impact.

Comments on the Quality of English Language

See above, it's fine.

Author Response

Minor comments:

line 38: "(e.g., cardiovascular diseases, ...)

line 40: "cell [4] and (b) the ..." (spacer is missing)

line 47: reference missing

line 54: reference is missing

line 89-91: Sentence lacks words... "are most abandoned on the WHAT?" and do the authors mean "detection limit" instead of "detection level"?

line 124: I think it should be "flow-induced".

All typos indicated by the Reviewer were corrected, and missing references were added to the revised text.

Major comments:

  1. The authors used only RBCs from male volunteers. Is there a difference between RBCs and women?

 The present study was confined to male subjects for a homogenous and stable sampling, as much as possible (as female donors are affected by their period). Accordingly, we collected blood samples from young (18-40 y.o.) healthy, non-anemic (Hb > 13 mg/dL) men with blood group type O+.

The revised manuscript adds a relevant note to the Methods (Lines 251 and 252) and Limitations (Lines 242 and 243) sections.

  1. Is there a difference between different blood groups?

As noted above, this study employed only samples with blood type O+ to assure the homogeneity of the donor population. To our knowledge, a difference in RBC deformability between blood types has not been reported.

  1. To be able to systematically analyze the described issues, I think that one need to characterize the RBC deformability in a Hb concentration dependent manner - even with the subunits. Are the authors able to provide such analysis for at least HBA and HBB at once? This would serve as a defined proof of your study and would increase the impact.

The method of mass spectroscopy used in the present study provides relative values. Their translation to weight values requires several assumptions and calculations, and it is conventional to express the results in the mass spec units [1,2]

We sincerely thank the Reviewer for his thorough review and constructive comments.

  1. Barshtein, G.; Gural, A.; Arbell, D.; Barkan, R.; Livshits, L.; Pajic-Lijakovic, I.; Yedgar, S. Red Blood Cell Deformability Is Expressed by a Set of Interrelated Membrane Proteins. Int J Mol Sci 2023, 24, doi:10.3390/ijms241612755.
  2. Geyer, P.E.; Wewer Albrechtsen, N.J.; Tyanova, S.; Grassl, N.; Iepsen, E.W.; Lundgren, J.; Madsbad, S.; Holst, J.J.; Torekov, S.S.; Mann, M. Proteomics reveals the effects of sustained weight loss on the human plasma proteome. Mol Syst Biol 2016, 12, 901, doi:10.15252/msb.20167357.

Reviewer 2 Report

Comments and Suggestions for Authors

The manuscript is certainly interesting, but several experimental controls are lacking. The oxidative status of  the Hb samples highly influences the degree of membrane binding.  The metHb and Fe(IV) forms bind more strongly and the levels of these need to be followed and controlled. In the present state, the manucript therefore provides very limited new knowledge to the field.

Comments on the Quality of English Language

The quality of English is fine.

Author Response

The Reviewer's comment is certainly interesting. MetHb is a form of Hb wherein its heme iron has been oxidized, changing its iron configuration from the ferrous (Fe2+) to the ferric (Fe3+) state. Usually, the MetHb content varies from 1 to 2% of the cytosol Hb. In some pathological states, the MetHb level might rise markedly (Jenni et al.) and affect the membrane state significantly. In the present study, we used the blood of young (18 - 40 years old), healthy, non-anemic donors, and it is unlikely that the MetHb content in the cytosol would exceed 2% of the total hemoglobin, which is insignificant compared to the total Hb content. 

As stated in the manuscript, the present study shows that for the first time

1 . a clear inverse correlation between the deformability of adults’ RBC and the level of the three main hemoglobin subunits that bind to the inner surface of the cell membrane - β, α, and δ. 

  1. a strong inter-correlation between the levels of these membrane-bound subunits
  2. the binding of the different hemoglobin isoforms to erythrocyte membranes is not competitive.

These new findings deserve publication in IJMS.

Jenni, S.; Ludwig-Peisker, O.; Jagannathan, V.; Lapsina, S.; Stirn, M.; Hofmann-Lehmann, R.; Bogdanov, N.; Schetle, N.; Giger, U.; Leeb, T., et al. Methemoglobinemia, Increased Deformability and Reduced Membrane Stability of Red Blood Cells in a Cat with a CYB5R3 Splice Defect. Cells 2023, 12, doi:10.3390/cells12070991.